# Prenylcysteine Oxidase 1 (PCYOX1), a New Player in Thrombosis

**DOI:** 10.3390/ijms23052831

**Published:** 2022-03-04

**Authors:** Cristina Banfi, Patrizia Amadio, Marta Zarà, Maura Brioschi, Leonardo Sandrini, Silvia S. Barbieri

**Affiliations:** 1Unit of Functional Proteomics, Metabolomics and Network Analysis, Centro Cardiologico Monzino IRCCS, 20138 Milano, Italy; cristina.banfi@ccfm.it (C.B.); maura.brioschi@ccfm.it (M.B.); 2Unit of Brain-Heart Axis, Centro Cardiologico Monzino IRCCS, 20138 Milano, Italy; patrizia.amadio@ccfm.it (P.A.); marta.zara@ccfm.it (M.Z.); leonardo.sandrini@ccfm.it (L.S.)

**Keywords:** Prenylcysteine Oxidase 1, thrombosis, platelets

## Abstract

Prenylcysteine Oxidase 1 (PCYOX1) is an enzyme involved in the degradation of prenylated proteins. It is expressed in different tissues including vascular and blood cells. We recently showed that the secretome from *Pcyox1*-silenced cells reduced platelet adhesion both to fibrinogen and endothelial cells, suggesting a potential contribution of PCYOX1 into thrombus formation. Here, we show that in vivo thrombus formation after FeCl_3_ injury of the carotid artery was delayed in Pcyox1^−/−^ mice, which were also protected from collagen/epinephrine induced thromboembolism. The Pcyox1^−/−^ mice displayed normal blood cells count, vascular procoagulant activity and plasma fibrinogen levels. Deletion of *Pcyox1* reduced the platelet/leukocyte aggregates in whole blood, as well as the platelet aggregation, the alpha granules release, and the α_IIb_β_3_ integrin activation in platelet-rich plasma, in response to adenosine diphosphate (ADP) or thrombin receptor agonist peptide (TRAP). Washed platelets from the Pcyox1^−/−^ and WT animals showed similar phosphorylation pathway activation, adhesion ability and aggregation. The presence of Pcyox1^−/−^ plasma impaired agonist-induced WT platelet aggregation. Our findings show that the absence of PCYOX1 results in platelet hypo-reactivity and impaired arterial thrombosis, and indicates that PCYOX1 could be a novel target for antithrombotic drugs.

## 1. Introduction

Despite the great advances made in the treatment of coronary artery disease (CAD), it is still the largest cause of death in industrialised and non-industrialised countries. The pathological mechanisms leading to CAD are various, but mainly related to inflammation and thrombosis. Specifically, thrombosis results from a complex interplay among vascular and blood cells, including platelets, red blood cells and leukocytes, and blood proteins (e.g., fibrinogen).

Platelets masterfully orchestrate the critical steps of haemostasis. At the sites of vessel injury, they adhere and aggregate to stem blood loss facilitating blood coagulation [1]. Through the same mechanisms, platelets are actively involved in thrombosis and in addition, by interacting with leukocytes, they also participate in the initiation and progression of atherosclerotic plaque [2]. Another important process is related to the reactive oxygen species (ROS), that by increasing the oxidative stress, induce the oxidation of lipoproteins, the activation of tissue factor (TF), the primary initiator of coagulation cascade, and enhance endothelial dysfunction and platelet hyper-reactivity. Platelets are not only the target of ROS but are themselves the source of ROS, which in turn, by sustaining platelet aggregation, promote, support, and amplify thrombus formation and growth [3,4].

However, despite the advance in understanding the mechanism underlying thrombosis and the critical role of platelets in this context, challenges in its prevention and treatment persist.

Prenylcysteine Oxidase 1 (PCYOX1) is an enzyme involved in the degradation of prenylated proteins, and in the release of hydrogen peroxide [5,6]. However, experimental and bioinformatics data [7,8] have showed that PCYOX1 might play additional roles beyond its involvement in the metabolism of prenylated proteins.

In addition, we recently showed that *Pcyox1* deficiency retards atheroprogression in Apoe^−/−^ mice, and it is associated with decreased features of lesion vulnerability. Finally, the secretome from *Pcyox1*-silenced cells reduced platelet adhesion both to fibrinogen and endothelial cells [7], suggesting a potential contribution of PCYOX1 into thrombus formation.

Given these premises, taking advantage of the *Pcyox1* gene knockout (Pcyox1^−/−^) C57BL/6J mice, here we investigated the effect of PCYOX1 deletion on thrombosis.

## 2. Results

### 2.1. Pcyox1 Deficiency Affects Thrombosis In Vivo

Prenylcysteine Oxidase 1 knock-out (Pcyox1^−/−^) mice display normal morphotype with body weight (WT: 26.37 ± 0.60 gr and Pcyox1^−/−^: 26.04 ± 0.52 gr, *p* = 0.673) and blood cell count (Table 1) similar to the littermate wild type (WT) mice.

To investigate the physiological relevance of PCYOX1 in arterial thrombosis, we tested the ability to form stable clots using a chemical-induced carotid injury model. The topical application of FeCl_3_ to the carotid artery rapidly induced occlusive thrombi that were stable over the 30 min length of the experiment in WT mice (Figure 1a,b). The Pcyox1 deficiency did not affect the basal blood flow (WT: 0.922 ± 0.052 mL/min and Pcyox1^−/−^: 0.846 ± 0.031 mL/min, *p* = 0.209), but completely prevented the carotid artery occlusion, and in almost all Pcyox1^−/−^ mice the thrombus completely failed to form (Figure 1a,b).

The in vivo significance of PCYOX1 in thrombosis was further confirmed by thromboembolism assay showing that the Pcyox1^−/−^ mice were also protected against pulmonary thromboembolism compared to the WT mice (Figure 1b). In particular, 61.5% of Pcyox1^−/−^ mice survived (5 mice died out of 13 tested) after a collagen/epinephrine injection compared to only 15.4% of WT mice (11 mice died out of 13 tested) (*p* < 0.005; Figure 1c).

These data clearly show that the *Pcyox1* deficiency protects mice from arterial thrombosis.

### 2.2. Impact of Pcyox1 Deletion on the Vascular Procoagulant Activity and Platelet Functions

Arterial thrombosis results from a complex interplay among vascular and blood cells. The vascular contribution of thrombosis has been assessed evaluating the TF activity, as procoagulant activity (PCA), in carotid artery from WT and Pcyox1^−/−^ mice. We found that the protective effect of *Pcyox1* deletion on thrombosis seems to be not related to alterations in the vascular thrombogenicity, as suggested by the lack of difference in TF procoagulant activity between Pcyox1^−/−^ and WT mice (Figure 2, *p* = 0.847).

Thus, in order to explain the defective thrombus formation in Pcyox1^−/−^ mice, we next investigated the potential contribution of blood cells in this context, focusing on platelet functions.

Firstly, we observed that the percentage of platelet/leukocyte aggregates in whole blood were similar in the Pcyox1^−/−^ and the WT mice at the basal condition, but were significantly reduced in Pcyox1^−/−^ mice upon stimulation with different concentrations of ADP (Figure 3a).

Then, we explored whether *Pcyox1* deficiency affected platelet alpha granule secretion and integrin αIIbβ3 (GPIIbIIIa) activation by evaluating P-selectin expression and JON/A binding, respectively. Platelet-rich plasma (PRP) analysis showed that platelets from Pcyox1^−/−^ mice have a reduced expression of P-selectin (Figure 3b), as well as diminished activation of GPIIbIIIa in response to adenosine diphosphate (ADP) and Thrombin Receptor Activating Peptide (TRAP) (Figure 3c).

Light transmission aggregometry on PRP using a range of dose of ADP (0.5–1 µM) and TRAP (0.6–2.5 µM) was performed to investigate the effect of *Pcyox1* deficiency on platelet aggregation.

Interestingly, the Pcyox1^−/−^ mice had marked impairment in PRP aggregation in response to both stimuli (Figure 4a–d). In particular, a consistent reduced maximum aggregation (ADP: −45%, and TRAP: −28% Pcyox1^−/−^ versus WT) and a lower area under the curve (AUC) [ADP: −70% and TRAP: −31% Pcyox1^−/−^ versus WT) was observed using a subthreshold concentration of both ADP (0.5 µM) and TRAP (0.6 µM) (Figure 4). The response was only partially restored raising the concentrations of both agonists (maximum aggregation: ~ −24%, and AUC: ~ −30%, for both 1 µM ADP and 1.25 µM TRAP) (Figure 4). When maximal aggregation was induced by the highest concentration of TRAP (2.5 µM), no more differences were observed between the two groups (maximum aggregation: WT: 91.5 % ± 3.7 versus Pcyox1^−/−^: 87.75 % ± 3.2, *p* = 0.457; AUC: WT: 363.7 ± 23.2 versus Pcyox1^−^^/^^−^: 336.4 ± 10.8, *p* = 0.342).

Overall, these data support the view that the absence of *Pcyox1* impairs platelet functions.

### 2.3. Role of PCYOX1 in Washed Platelets

To understand the relevance of PCYOX1 on platelet activation, we firstly provide evidence of the presence of PCYOX1 in washed platelets isolated from WT mice (0.89 ± 0.14 amol PCYOX/ng total proteins) assessed by quantitative mass spectrometry analysis (Figure 5), and then we evaluated platelet functions in washed platelets in response to TRAP.

Interestingly, when aggregation was carried out in washed platelets, we did not observe any statistical difference between the two groups. Specifically, the aggregation curves of washed platelets isolated from Pcyox1^−/−^ mice stimulated with TRAP were perfectly comparable to those isolated from WT mice (Figure 6).

Then we evaluated the effect of *Pcyox1* deletion on platelet adhesion and spreading on surfaces coated with fibrinogen (100 μg/mL) by fluorescence microscopy method.

We failed to observe significant differences in the adhesion to fibrinogen matrix between the two groups during 40 min observation time (Figure 7a,b). The surface area occupied by each platelet as well as the number of spread platelets isolated from Pcyox1^−/−^ mice was similar to those isolated from WT mice (Figure 7c,d). Similar results have been obtained when washed platelets were stimulated with TRAP (data not shown).

Finally, the normal phenotype of platelet lacking *Pcyox1* was additionally confirmed by the analysis of specific phospho-protein-pathways involved in platelet activation. Indeed, the absence of *Pcyox1* did not alter the phosphorylation kinetics of p38 mitogen-activated protein kinase (p38MAPK), extracellular signal-regulated kinase 1/2 (Erk1/2), and protein kinase B (Akt) after stimulation with TRAP in washed platelets (Figure 8a–e).

Overall, these data suggest that the absence of *Pcyox1* in platelets did not affect their activation.

### 2.4. Role of Plasma Component in the Modulation of Platelet Activation in Pcyox1 Deficient Mice

Once it was established that impaired aggregation in the Pcyox1^−/−^ mice was independent of intrinsic platelet *Pcyox1* deletion, we evaluated the potential contribution of circulating factors present in the Pcyox1^−/−^ plasma by performing aggregation analysis using washed WT platelets in presence of plasma isolated from WT or Pcyox1^−/−^ mice.

A reduced platelet aggregation in response to ADP was observed when washed WT platelets were resuspended in Pcyox1^−/−^ plasma than in WT plasma, with a reduction of the maximum aggregation and AUC to 53% and 37%, respectively (Figure 9).

However, this effect was independently confirmed by circulating fibrinogen levels, which were comparable between the two groups of mice (Figure 10, *p* = 0.733).

## 3. Discussion

In this study we have shown for the first time that PCYOX1 is a critical modulator of arterial thrombosis as provided by the impressive low thrombotic phenotype here observed in Pcyox1^−/−^ mouse under both models of thrombosis. Specifically, we found that PCYOX1 is present in murine platelets, and that its absence impairs platelets activation reducing the platelet/leukocyte aggregates in whole blood, as well as the platelet aggregation, the alpha granules release, and the α_IIb_β_3_ integrin activation in platelet-rich plasma. However, as washed platelets from Pcoxy1^−/−^ and WT animals show similar phosphorylation pathway activation, adhesion ability, and aggregation, it follows that, as suggested by cross-plasma platelet aggregation experiments, the loss of PCYOX1 might influence indirectly platelet behaviour by affecting plasma composition. The significance and the function of PCYOX1 presence in platelets remain completely unknown.

The component(s) responsible for the effect here observed remains to be identified. Currently, we can exclude that the protective effect of *Pcyox1* deletion on thrombosis is related to vascular alteration, as suggested by there being no difference in aorta tissue procoagulant activity between the Pcyox1^−/−^ and WT mice. However, future studies focused on a detailed characterization of the vascular phenotype of the Pcyox1^−/−^ mouse are needed. We can simply hypothesize that the presence of PCYOX1 might affect functions and/or composition of soluble factors or macromolecules, such as lipoproteins, that play a role in arterial thrombosis.

PCYOX1 has recently emerged as a lipoprotein-associated protein, representing a potential novel member in the panel of oxidant enzymes involved in the pathogenesis of atherosclerosis [7]. Its biological functions have thus extended from the former findings by Casey’s group who described PCYOX1 as a lysosomal enzyme involved in the catabolism of prenylated proteins [9]. PCYOX1 is a flavin adenine dinucleotide (FAD)-dependent thioether oxidase that produces free cysteine, an isoprenoid aldehyde, and a stoichiometric amount of hydrogen peroxide [10]. Our previous study identified PCYOX1 as a novel member in the panel of oxidant enzymes involved in the pathogenesis of atherothrombosis [7]. Being a FAD-dependent thioether oxidase distinguishes it from other oxidant enzymes. Indeed, the known ROS–producing systems in the vascular wall currently include Nicotinamide-adenine dinucleotide phosphate (NADPH) oxidase, xanthine oxidase, the mitochondrial electron transport chain, and uncoupled endothelial nitric oxide (NO) synthase. Further, unlike these enzymes, which are all intracellular or endothelium-bound, as in the case of xanthine oxidase, PCYOX1 is also a circulating oxidant enzyme with a role in atherogenesis.

Of note, we previously demonstrated that PCYOX1 is bound to apolipoprotein B (apoB)-enriched lipoproteins and it can produce H_2_O_2_, which may directly contribute to platelet response [11], enhance thrombotic susceptibility [12], or in turn oxidize the apoB100-containing lipoprotein itself supporting platelet activation [13]. We showed, for example, that the activation of PCYOX1 bound to lipoprotein increased in vitro the procoagulant activity of Tissue Factor (TF), a gene modulated by oxidant species [14], in human aortic endothelial cells. Indeed, the accumulation of oxidized lipids in the plasma of high-risk patients is associated with increased platelet reactivity [15], whereas oxidized low-density lipoproteins (oxLDLs), both generated in vitro and isolated from subjects with cardiovascular disease (CVD), can contribute to platelet activation, suggesting them as potential causative agents promoting platelet hyperactivity in CVD [16,17].

Also, soluble factors might affect platelet functions. Remarkably, we recently demonstrated that the secretome of *Pcyox1*-silenced cells (HepG2 cells) impaired in vitro adhesion of human platelets to both fibrinogen-coated plates and human endothelial cells. Further, the secretome analysis revealed that PCYOX1 is a multifunctional protein potentially involved in a plethora of systems including the regulation of peptidase activity, platelet degranulation, regulation of signal transduction, response to stress, regulation of response to stimulus, inflammatory response, and response to wounding [7]. We also showed that Pcyox1^−/−^/Apoe^−/−^ mice have a significant decrease in plasma activity of PAI-1, the primary physiological inhibitor of fibrinolysis [18].

Interestingly, several lines of evidence indicate that platelet/leukocyte aggregates contribute not only to thrombosis but also to the inflammatory process [19]. The lower percentage of platelet/leukocyte aggregates measured in Pcyox1^−/−^ mice compared to WT mice may then reflect a reduced inflammatory state as suggested by our previous study where macrophages isolated from mice lacking *Pcyox1* synthesized a lower amount of pro-inflammatory cytokines compared to WT [7]. Whether PCYOX1 is a link between inflammation and thrombosis must be investigated in the future. Despite these unsolved questions, this study supports the concept that PCYOX1 represents a novel player in the complex scenario of atherothrombosis.

## 4. Materials and Methods

### 4.1. Mice

All experiments were performed in adult WT and littermate Pcyox1^−/−^ mice (3–4 months old) derived from Pcyox1^+/−^ × Pcyox1^+/−^ mice [7], and were genotyped by PCR analysis. Mice were housed under standard conditions (20–22 °C, 12 h light/dark cycle) with water and food ad libitum. All animal procedures were conformed to the rules and principles of the 2010/63/EU Directive, approved, and authorized by the National Ministry of Health-University of Milan Committee and Cogentech S.r.l.

Surgical procedures were performed in mice anesthetized with ketamine chlorhydrate (75 mg/kg; Intervet, Milan, Italy) and medetomidine (1 mg/kg; Virbac, Milan, Italy).

### 4.2. Blood Collection, Blood Cell Count and Plasma Preparation

Blood was collected into 3.8% sodium citrate (1:10 vol:vol) by cardiac venipuncture from anesthetized mice, and platelets as well as white and red blood cells were immediately counted optically [20].

For plasma preparation anticoagulated blood was centrifuged at 2000× *g* for 20 min at 4 °C and stored at −80 °C until further analysis.

### 4.3. Carotid Artery Thrombosis Model

The FeCl_3_ injury of the carotid artery was performed in anesthetized mice as previously described [21]. Briefly, after the left carotid artery was dissected free, the blood flow was monitored with a Doppler flow probe (model 0.7 V, Transonic System, Transonicm, Ithaca, NY, USA) connected to a transonic flow meter (Transonic T106). When the blood flow was stable and constant for 7 min at least 0.8 mL/sec, a 1 × 1 mm strip of Whatman N°1 filter paper soaked with 10% FeCl_3_ (Sigma-Aldrich, St. Louis, MO, USA) was placed over the carotid artery for 3 min. Then, the flow was recorded for 30 min after removing the filter paper and washing the carotid artery with phosphate buffered saline (PBS). An occlusion was considered to be total when the flow was reduced by more than 90% of baseline within and during 5 min.

### 4.4. Mouse Thromboembolism Model

A disseminated thrombosis model was generated as previously described [22] by the intravenous injection of a mixture of collagen (845 μg/kg, Mascia Brunelli, Milan, Italy) and epinephrine (60 μM/kg Sigma-Aldrich, St. Louis, MO, USA). Mice were observed for 15 min and the time of cessation of breathing stasis recorded.

### 4.5. Procoagulant Activity

The carotid arteries excised from the WT and Pcyox1^−/−^ mice were cut, resuspended in 150 µL of 15 mM β-octil-glucopiranoside (Sigma-Aldrich, St. Louis, MO, USA), sonicated for 30 min and held at 37 °C for 15 min before performing procoagulant activity assay (PCA). After specific protein quantification (Bradford method), samples were conveniently diluted with HEPES to obtain final protein concentration of 1.25 µg/40 µL. The PCA was measured by recalcification time test of citrated mouse plasma pool at 37 °C. To 40 µL of sample, pre-incubated at 37 °C in temperature bath for 1 min, were added 40 µL of platelet-poor plasma (PPP) and 40 µL of 15 mM CaCl_2_. Recalcification times have been converted to arbitrary units of tissue factor (TF) by using a calibration curve of thromboplastin extracted from human placenta (CSL Behring, Milan, Italy) [23].

### 4.6. Platelet-Rich Plasma Preparation

Blood collected into 3.8% sodium citrate was then diluted 1:1 with Hepes–Tyrode’s buffer (137 mM NaCl, 20 mM HEPES, 5.6 mM glucose, 0.35% bovine serum (BSA), 1 mM MgCl_2_, 2.7 mM KCl, 3.3 mM NaH_2_PO_4_) and centrifuged at 100× *g* for 10 min. Supernatant, platelet rich plasma (PRP), was gently removed with a plastic pipette and platelet counts were carried out in a Burker chamber after 1:200 dilution of the sample into Hepes–Tyrode’s buffer. PPP was obtained by centrifugation of the remaining blood at 2000 g for 20 min. Platelet counts were adjusted to 250 × 10^3^ platelets/µL with autologous PPP.

### 4.7. Platelet Flow Cytometry Analyses

Platelet-leukocyte aggregate analyses were performed in citrated blood stimulated for 5 min with Adenosine diphosphate (ADP). Then, after the lyses of red blood by FACS lysing solution (BD Biosciences, San Jose, CA, USA), samples were centrifuged, stained with anti-CD45 and anti-CD-61 (BD Biosciences, San Jose, CA, USA), and analysed by flow FACS “Novocyte 3000”. A minimum of 10,000 events was collected in the CD45^+^ gate [24].

Platelet activation was assessed in PRP using an anti-JON/A (Emfret Analytics, Elbestadt, Germany) or anti-CD62P (P-selectin; BD Biosciences, San Jose, CA, USA) antibody. Twenty-five µL of PRP were mixed with a saturating concentration of antibody and the mixture reacted with different concentrations of ADP or Thrombin Receptor Agonist Peptide (TRAP) for 15 min at room temperature. The reaction was stopped by 500 µL ice-cold PBS, and samples were analysed within 30 min. A minimum of 10,000 events was collected [25].

### 4.8. Washed Platelet Preparation

Washed platelets were prepared from blood collected using ACD/3.8% sodium citrate (2:1) as anticoagulant as previously described [26]. Briefly, anticoagulated blood was diluted with HEPES buffer (10 mM HEPES, 137 mM NaCl, 2.9 mM KCl, 12 mM NaHCO3, pH 7.4) and centrifuged for 10 min at 180× *g* to obtain platelet-rich-plasma (PRP). The PRP was then transferred to new tubes and the remaining red blood cells were diluted with HEPES buffer and centrifuged again at 180× *g* for 7 min. The upper phase was added to the previously collected PRP and 0.02 U/mL apyrase, and 1 μM prostaglandin E_1_ (PGE_1_) were added before centrifugation at 550× *g* for 10 min. The supernatant was removed, and the platelet pellet was washed in PIPES buffer (20 mM PIPES, 136 mM NaCl, pH 6.5) and centrifuged at 720× *g* for 15 min. The platelet pellet was finally gently resuspended in HEPES buffer and counted. The platelet count was adjusted in HEPES with 5.5 mM glucose, 1mM CaCl_2_, 0.5 mM MgCl_2_ or PPP at a final concentration of 250 × 10^3^ platelets/µL. Platelets were allowed to rest for 30 min at room temperature before use.

### 4.9. Platelet Aggregation Studies

Platelet aggregation was assessed using light transmission aggregometry (Chrono-Log, Havertown, PA, USA) in PRP, WPs or WPs plus PPP under 1000 rpm constant stirring at 37 °C. Aggregation was initiated by the addition of various concentrations of ADP (0.5–1 µM) or TRAP (0.6–2.5 µM) as indicated and tracings were recorded for 5 min. Area under the curve (AUC) and % maximum aggregation (% Max Agg) were calculated [27].

### 4.10. Western Blot

Washed platelets were lysed in cold RIPA buffer (62.5 mM TRIS HCl, 100 mM NaCl, 1% NP-40, 0.1% 13 Tween 20, 1 mM Na-orthovanadate, 1 mM PMSF, 10 mM Na-pyrophosphate, 10 mM NaF 14 pH 8.0 and protease inhibitor cocktail), sonicated and protein yield was quantified using the BCA protein assay kit (Sigma-Aldrich, St. Louis, MO, USA). Twenty micrograms of protein samples were prepared with the Laemmli method, and equivalent amounts of protein were separated on 12% SDS-PAGE gels, transferred to nitrocellulose membrane and bands of interest detected using rabbit anti-phospho p38, mouse anti-phospho ERK1/2 and rabbit anti-phospho Akt (1:1000; Cell Signalling technology, Danvers, MA, USA). Then membranes were incubated with peroxidase-conjugated secondary antibody (goat anti-rabbit from BioRad laboratories, Milan, Italy and anti-mouse from Sigma-Aldrich, St. Louis, MO, USA). Immunoreactive bands were visualized by enhanced chemiluminescence (GE healthcare, Chicago, IL, USA) and analysed with the QuantityOne software (Bio-Rad Laboratories, Milan, Italy) for densitometric analysis including normalization for total protein loading visualized with MEMcode (Bio-Rad Laboratories, Milan, Italy) [28].

### 4.11. Platelet Adhesion

Coverlips (10 mm Ø) were coated with 100 µg/mL fibrinogen (Sigma-Aldrich, St. Louis, MO, USA), diluted in PBS Ca^2+/^Mg^2+^ o/n at 4 °C. Subsequently, the coverlips were blocked with 5% BSA for 1 h. Then 3 × 10^4^ washed platelets were added to fibrinogen-coated coverlips and allowed to adhere for the indicated time at 37 °C in the presence of only 1 mM CaCl_2_. Wells were then washed with PBS to remove non-adherent platelets, and adherent platelets were fixed with 4% paraformaldehyde for 10 min, permeabilized in 0.2% Triton X-100 for 5 min, washed with PBS, blocked with 5% BSA for 1 h and stained with Alexafluor 488-phalloidin (1:200, Invitrogen, Waltham, MA, USA) for 1 h at room temperature. Images of coverlips were acquired by Apotome microscope (Carl Zeiss, Jena, Germany). The number of adherent cells (digital images 20×) was counted and the surface areas (digital images 40×) were measured in 6 different fields for each sample using Image J software (National Institutes of Health, Bethesda, MD, USA) [22].

### 4.12. Functional Fibrinogen Assay

The concentration of functional fibrinogen in plasma was measured by the Clauss method as previously described [29].

### 4.13. Mass Spectrometry Analysis

For proteomic analysis, platelets were dissolved in 25 mmol/L NH_4_HCO_3_ containing 0.1% RapiGest (Waters Corporation, Milford, MA, USA), sonicated, and centrifuged at 13,000× *g* for 10 min. Samples (50 μg of protein) were then incubated for 15 min at 80 °C and reduced with 5 mmol/L DTT at 60 °C for 15 min, followed by carbamidomethylation with 10 mmol/L iodoacetamide for 30 min at room temperature in the darkness. Then, 2.5 μg of sequencing grade trypsin (Promega) was added to each sample and incubated overnight at 37 °C. After digestion, 2% TFA was added to hydrolyse RapiGest and inactivate trypsin. Tryptic peptides (0.5 ug/ul) with the addition of stable isotope labelled proteotipic PCYOX1 peptide (CPSIILHD(R) from Thermofisher Scientific (Walthan, MA, USA)) were then desalted with ZipTip C18 (Millipore, Burlington, MA, USA) according to manufacturer instruction, and dissolved in water with 0.1% formic acid before mass spectrometry analysis.

Two microliters of each sample, containing 10 fmol/μL of labelled heavy peptide, were injected into a Xevo TQ-S micro triple quadrupole mass spectrometer coupled to a Waters ACQUITY ultra-performance liquid chromatography (UPLC) M-Class system through an ionKey source (Waters Corporation, Milford, MA, USA). The instrument was operated in positive ion mode in unit resolution. The capillary voltage was maintained at 3.20 kV with the source temperature held constant at 120 °C. After isocratic trapping for 1 min with a flow rate at 30 μL/min 99.5% solution A (99.9% LC-MS-grade water with 0.1% formic acid)/0.5% solution B (LC-MS-grade 99.9% acetonitrile with 0.1% formic acid) on a ACQUITY UPLC M-Class Symmetry C18 Trap Column, 100 Å, 5 μm, 300 μm × 50 mm (Waters Corporation, Milford, MA, USA) column, an iKey Peptide BEH C18 column, 130 Å 1.7 μm, 150 μm × 100 mm (Waters Corporation, Milford, MA, USA) was used for peptide separation. The flow rate was set to 3 μL/min with a column temperature of 60 °C. A gradient of solvent A and solvent B was applied with a total run time of 25 min as follows: 0–2.5 min at 2% B; 2.5–2.6 min from 2 to 5% B; 2.6–11 min linear increase from 5 to 23% B; 11–15 min from 23 to 30% B; 15–17 min from 30 to 45% B; 17.5–19.5 min 90% B; 20–25 min 2% B. Optimized collision energies for the selected transitions for multiple reaction monitoring (MRM) analysis are reported in Table 2. Peptide quantity, calculated from the ratio of the integrated area of the endogenous peptide’s transitions to the area of the corresponding heavy peptide’s transitions, was extrapolated from a standard curve generated with increasing concentrations of light peptide (0.125–5 fmol/µL) and fixed concentration of heavy peptide (5 fmol/µL). Skyline (v4.1) was used for area extraction and Excel for calculation.

To normalized PCYOX1 quantitation, label-free mass spectrometry analysis, LC-MS^E^, was performed to obtain a molar estimation of the total amount of proteins in respect to a known amount of alcohol dehydrogenase digest [30] on a hybrid quadrupole time-of-flight mass spectrometer (Synapt XS, Waters corporation, Milford, MA, USA) coupled with a UPLC Mclass system and equipped with a nanosource (Waters Corporation, Milford, MA, USA). Samples were injected into a Symmetry C18 nanoACQUITY trap column, 100 Å, 5 μm, 180 μm × 2 cm (Waters Corporation, Milford, MA, USA) and subsequently directed to the analytical column HSS T3 C18, 100 Å, 1.7 μm, 75 μm × 150 mm (Waters Corporation, Milford, MA, USA), and analysed by LC-MS^E^ as previously detailed [31]. The DIA data were processed and searched using ProteinLynx GlobalSERVER (PLGS) version 3.0 (Waters Corporation, Milford, MA, USA) using the mouse UniProt database (v2021-03).

### 4.14. Statistical Analyses

Statistical analyses were performed with GraphPad Prism v9.0 (La Jolla, CA, USA). Data are presented as mean ± standard error of the mean (SEM) and analysed by the Mann–Whitney test, or by two-way ANOVA with or without repeated measures followed by Bonferroni post hoc analysis by using log-transformed variables. In the Kaplan-Meier analysis, the comparison between survival curves was made by the Log-Rank test. Values of *p* < 0.05 were considered statistically significant.

## 5. Conclusions

In conclusion, this study demonstrates a critical role of Prenylcysteine Oxidase 1 in the regulation of thrombus formation and in the coordination of platelet function. Pcyox1^−/−^ mice had less susceptibility to thrombosis associated with a reduced platelet activation potentially related to the plasma composition. Although further studies are needed to understand which plasma component could be responsible of the different behaviour of platelets isolated from WT and Pcyox1^−/−^ mice, Prenylcysteine Oxidase 1 may represent a promising target in the prevention and treatment of arterial thrombosis.

## 6. Patents

We disclose the following European Patent application, under examination, EP15817414.4 entitled “Prenylcysteine Oxidase 1 inhibitors for the prevention and/or treatment of oxidative stress-related degenerative diseases and Prenylcysteine Oxidase 1 as diagnostic marker”, applicant Centro Cardiologico Monzino IRCCS, inventors C.B., M.B., R.B. and S.S.B., with no financial competing interest. The remaining authors declare no competing interest.

## Figures and Tables

**Figure 1 ijms-23-02831-f001:**
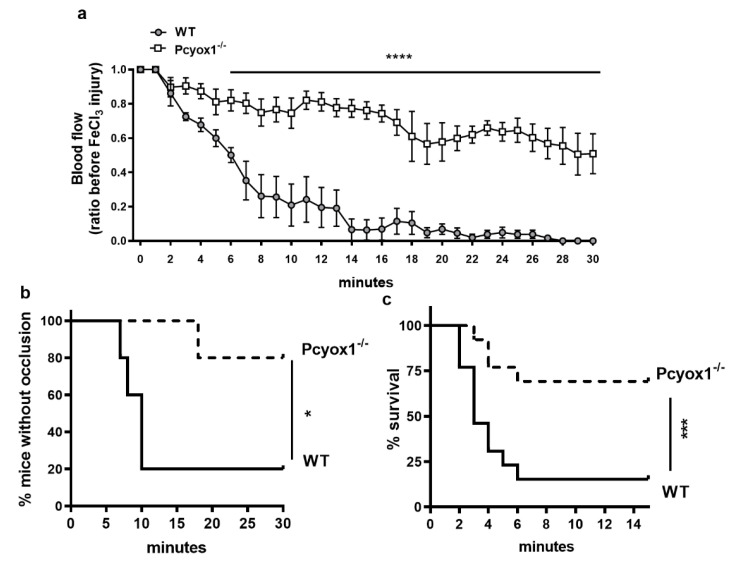
Thrombosis in WT and Pcyox1^−/−^ mice. (**a**,**b**) Arterial thrombosis induced by FeCl_3_ application to the carotid artery: (**a**) blood flow kinetic in carotid arteries during 30 min of observation was expressed relative to the value before the injury, and (**b**) Kaplan–Meier curve representing percentage of mice without occlusion through recording time. *n* = 5 mice/group. (**c**) Kaplan–Meier curve representing percentage of mice survived to collagen/epinephrine injection. *n* = 13 mice/group. (**a**) Data are shown as mean ± SEM. Two-way ANOVA with repeated measures followed by Bonferroni post hoc analysis *p* value were obtained by using log-transformed variables, and (**b**,**c**) *p*-values were obtained by analysing Kaplan–Meier curves by Log-Rank tests. * *p* < 0.05, *** *p* < 0.005 and **** *p* < 0.001.

**Figure 2 ijms-23-02831-f002:**
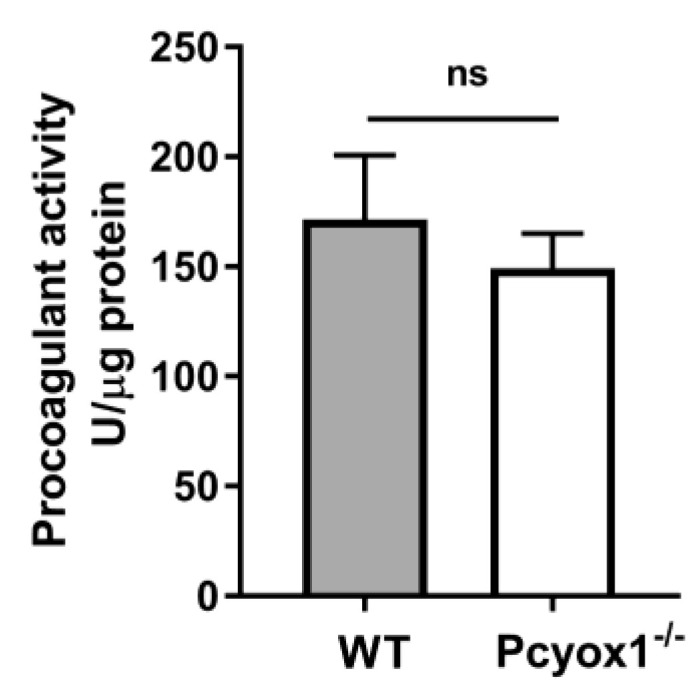
Procoagulant activity of WT and Pcyox1^−/−^ mice. Carotid artery procoagulant activity of WT and Pcyox1^−/−^ mice. *n* = 15 mice/group. Data are shown as mean ± SEM. ns = not significance at Mann-Whitney U-test.

**Figure 3 ijms-23-02831-f003:**
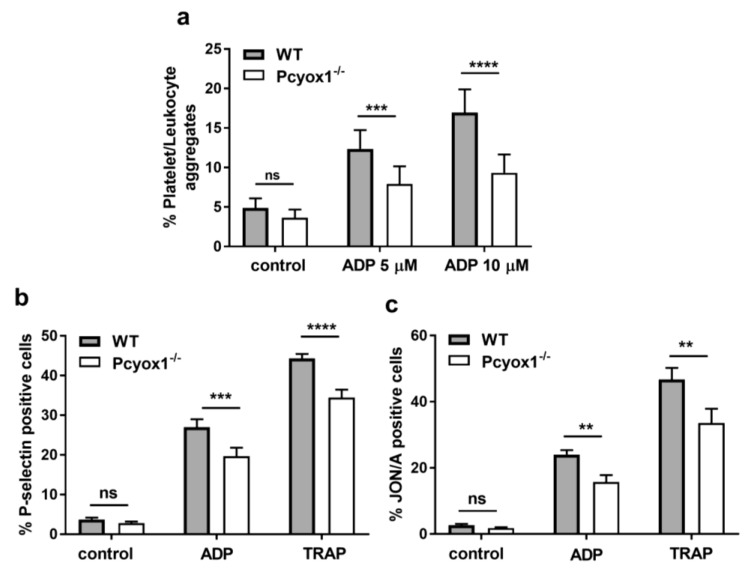
Analysis of platelet function in WT and Pcyox1^−/−^ mice. Flow cytometry analyses of: (**a**) platelet/leukocyte aggregates in the whole blood, in resting conditions and after stimulation with ADP; (**b**) expression of P-selectin, and (**c**) activation of integrin αIIbβ3 (GPIIbIIIa; (JON/A-PE antibody) in platelet rich plasma (PRP) at basal condition or after exposure to ADP (5 µM), or TRAP (6 µM). *n* = 6 mice/group. Data are shown as mean ± SEM. Two-way ANOVA followed by Bonferroni post hoc analysis *p*-value were obtained by using log-transformed variables. ns: non-significant, ** *p* < 0.01, *** *p* < 0.005 and **** *p* < 0.001.

**Figure 4 ijms-23-02831-f004:**
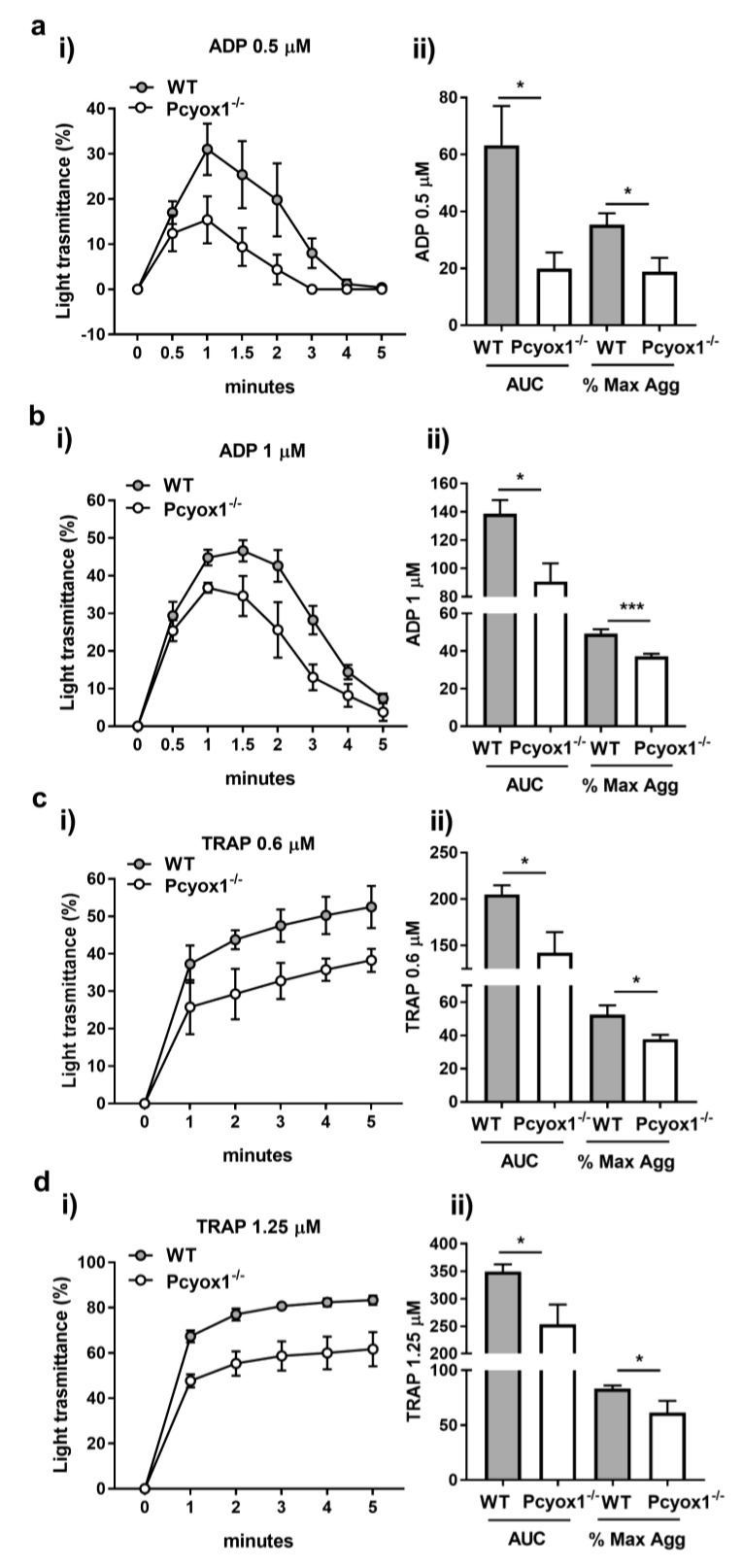
Platelet-rich plasma aggregation in WT and Pcyox1^−/−^ deficient mice. Platelet aggregation was performed in PRP in response to different concentrations of (**a**,**b**) ADP (0.5–1 µM) and (**c**,**d**) TRAP (0.6–1.25 µM). Data are shown as (i) kinetic of light transmission, and (ii) area under the curve (AUC) and % of maximum aggregation (% Max Agg) during the 5 min of recording. (Grey bar: WT mice; white bar: Pcyox1^−/−^ mice). *n* = 5 mice/group. Data are shown as mean ± SEM. * *p* < 0.05, and *** *p* < 0.005 at Mann-Whitney U-test.

**Figure 5 ijms-23-02831-f005:**
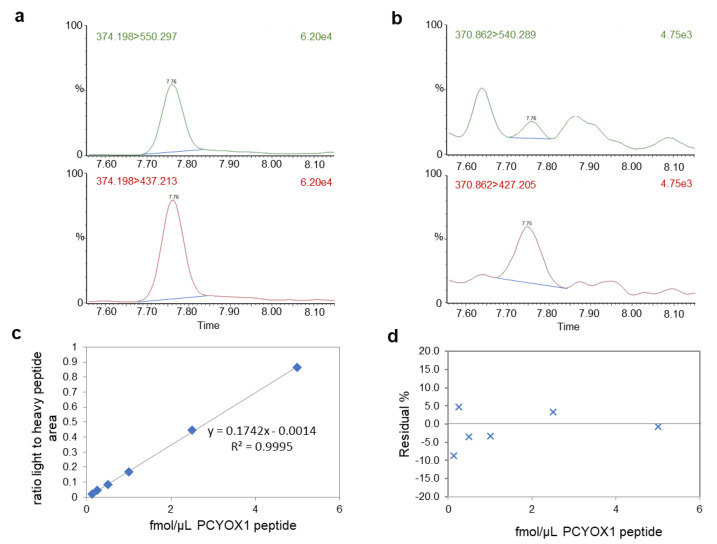
Analysis of the PCYOX1 protein in platelets from WT mice by mass spectrometry with a multiple reaction monitoring approach. Representative images of the chromatogram of the transitions of the heavy peptide (**a**) and the endogenous proteotipic peptide of PCYOX1 (**b**). (**c**) Standard curve generated with increasing concentrations of light peptide (0.125–5 fmol/µL) and fixed concentration of heavy peptide (5 fmol/µL). Data are expressed as the ratio of light peptide area versus heavy peptide area. The equation and the R^2^ value are reported. (**d**) Relative residual error in the extrapolation of light peptide concentration expressed as % of the expected peptide concentration. The PCYOX1 content was then normalized for total protein content measured by label-free mass spectrometry.

**Figure 6 ijms-23-02831-f006:**
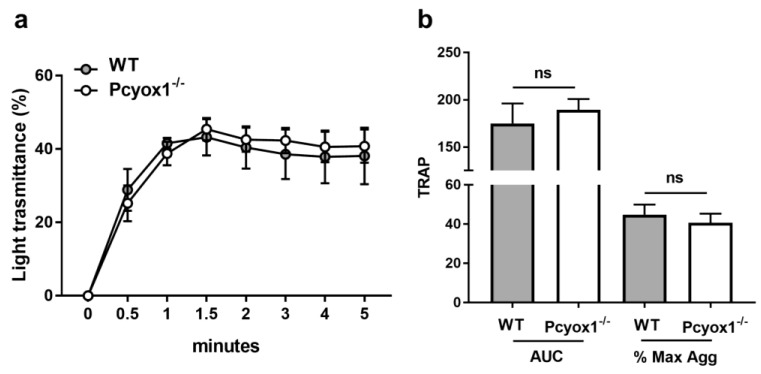
Aggregation of washed platelets isolated from the WT and Pcyox1^−/−^ mice. Platelet aggregation was performed in washed platelets in response to subthreshold concentration of TRAP. Data are shown as (**a**) kinetic of light transmission and (**b**) area under the curve (AUC) and % of maximum aggregation (% Max Agg) during the 5 min of recording. (Grey bar: WT mice; white bar: Pcyox1^−/−^ mice). *n* = 7 mice/group. Data are shown as mean ± SEM. ns = not significant at Mann–Whitney U-test.

**Figure 7 ijms-23-02831-f007:**
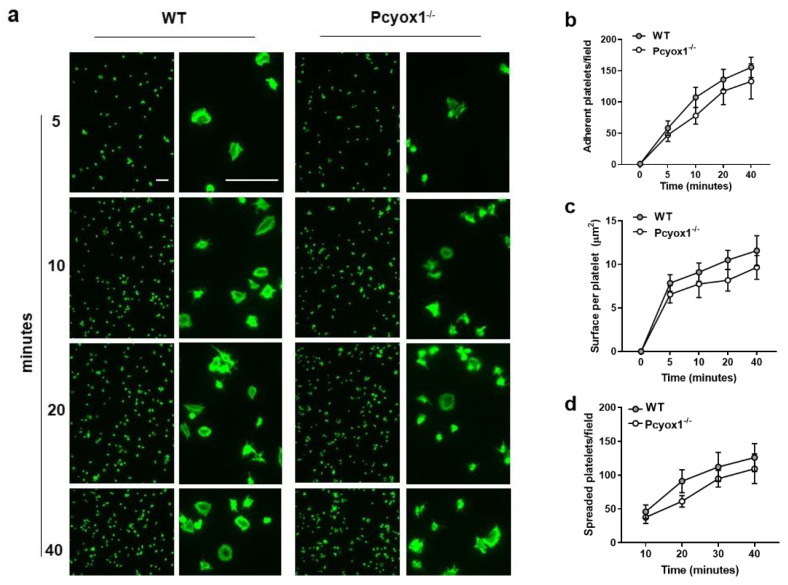
Effect of *Pcyox1* deletion on platelet adhesion isolated from the WT and Pcyox1^−/−^ mice. (**a**) Representative images of washed platelet adhesion on fibrinogen-coated surfaces. Scale bar 10 µm. Quantitation of WT (grey) and Pcyox1^−/−^ (white) washed platelets on fibrinogen matrix in terms of: (**b**) platelets number per field, (**c**) platelet area, and (**d**) number of spreaded platelets per field. *n* = 6 independent experiments/group. Data are shown as mean ± SEM. Two-way ANOVA followed by Bonferroni post hoc analysis *p*-value were obtained by using log-transformed variables.

**Figure 8 ijms-23-02831-f008:**
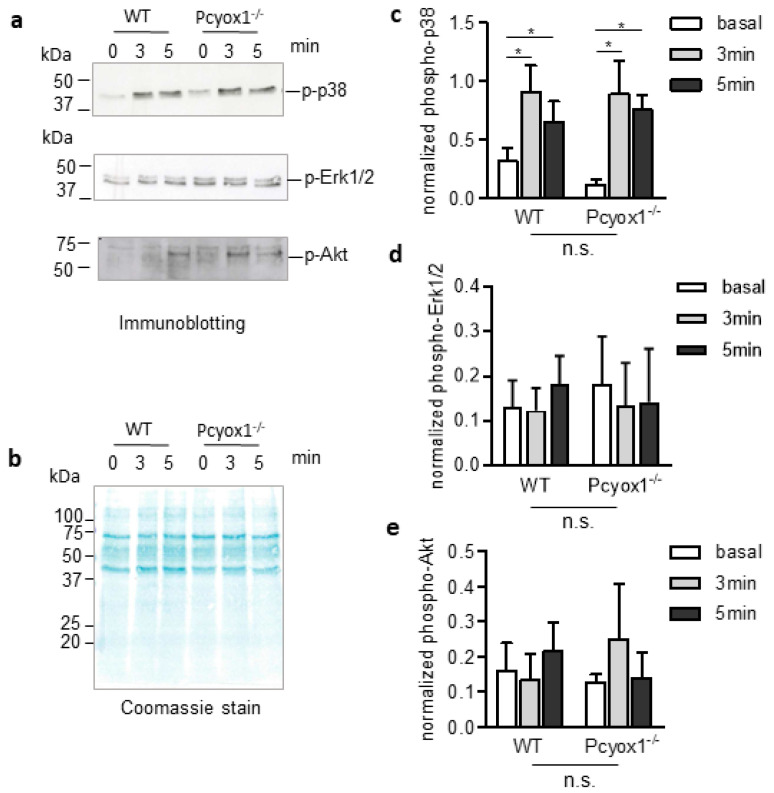
Characterization of phospho-protein-signalling pathways in washed platelet (WP) isolated from the WT and Pcyox1^−/−^ mice. The WPs isolated from the WT and Pcyox1^−/−^ mice were activated with TRAP for the indicated time. (**a**) Representative Western blot, (**b**) coomassie stain of total proteins, and quantification of kinetic of (**c**) phospho-p38MAPK (p-p38), (**d**) phospho-Erk1/2 (pErk/2), and (**e**) phospho-Akt. *n* = 4 independent experiments/group. Data are shown as mean ± SEM. Two-way ANOVA followed by Bonferroni post hoc analysis revealed a significant increase of phosphorylation of p38 with time (*p* = 0.0063), but no differences with genotype or interactions have been detected for p-p38, p-Erk1/2 and p-Akt. * *p* < 0.05 versus basal levels.

**Figure 9 ijms-23-02831-f009:**
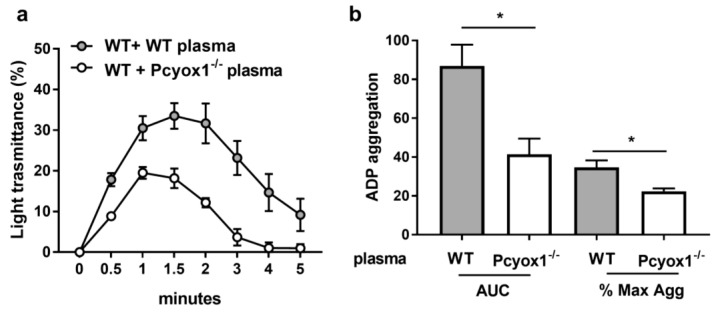
Aggregation of washed WT platelets in plasma isolated from the WT and Pcyox1^−/−^ mice. Platelet aggregation was performed in response to subthreshold concentration of ADP. Data are shown as (**a**) kinetic of light transmission, (**b**) area under the curve (AUC) and % of maximum aggregation (Max Agg) during the 5 min of recording. (Grey bar: WT plasma; white bar: Pcyox1^−/−^ plasma). Washed platelets per assay were pooled from 3 mice, WT or Pcyox1^−/−^ plasma per assay pooled from 2 mice. *n* = 6 independent experiments. Data are shown as mean ± SEM. * *p* < 0.05 at Mann–Whitney U-test.

**Figure 10 ijms-23-02831-f010:**
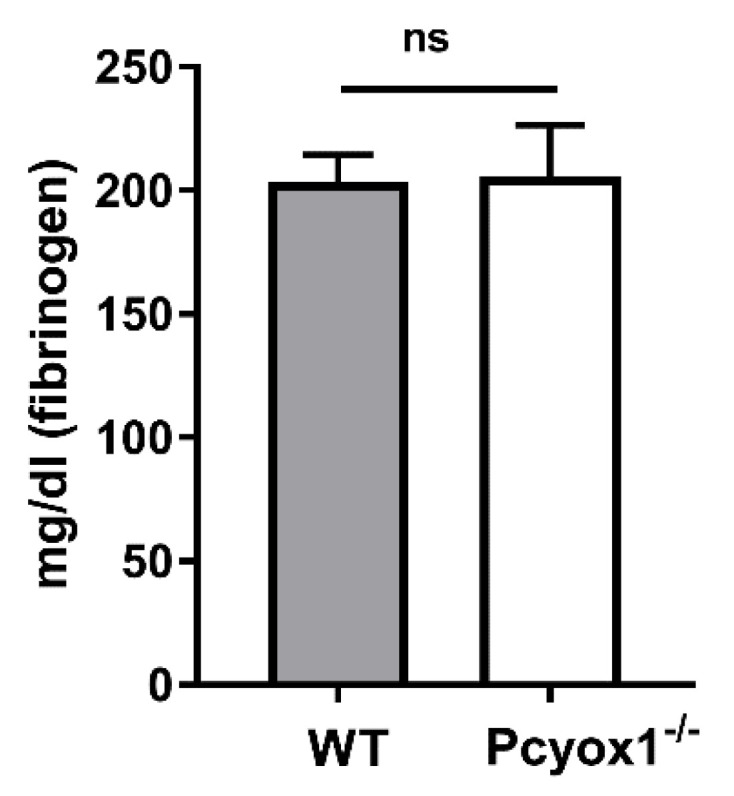
Fibrinogen levels in the WT and Pcyox1^−/−^ mice. Fibrinogen was detected by clauss methods in plasma of WT and Pcyox1^−/−^ mice. *n* = 15 mice/group. Data are shown as mean ± SEM. Ns = not significance at Mann-Whitney U-test.

**Table 1 ijms-23-02831-t001:** Blood cell count in WT and Pcyox1^−/−^ mice.

Cells	WT	Pcyox1^−/−^	*p*-Value
Erythrocytes (10^7^/µL)	15.47 ± 0.65	15.88 ± 0.99	0.650
Leukocytes (10^6^/µL)	2.61 ± 0.97	2.81 ± 1.57	0.301
Platelets (10^4^/µL)	71.9 ± 2.20	72.47 ± 2.61	0.707

Data are expressed as mean ± SEM. *n*: wild type (WT) = 16 mice and Prenylcysteine Oxidase 1 knock-out (Pcyox1^−/−^) = 10 mice.

**Table 2 ijms-23-02831-t002:** Optimized parameters for PCYOX1 analysis by mass spectrometry.

Peptide	Precursor *m*/*z*	Transition *m*/*z*	Collision Energy	Cone Voltage (V)
CPSIILHDR light	370.86	540.29	15	35
		427.20 *	15	35
CPSIILHDR heavy	374.2	550.2971	15	35
		437.2131 *	15	35

* quantifier transition.

## Data Availability

Data collected in the study will be made available using the data repository Zenodo (https://zenodo.org/) with restricted access upon request to direzione.scientifica@ccfm.it.

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
