# Peer review of "Prenylcysteine Oxidase 1 (PCYOX1), a New Player in Thrombosis"

_ijms, 2022, doi:10.3390/ijms23052831_

Round 1

Reviewer 1 Report

The findings of this study strongly support a novel function for the protein Prenylcysteine Oxidase 1 in thrombosis involving platelet hypo-reactivity.

The manuscript clearly outlines the strategy and rationale for the study. Its findings are clearly presented and argued, with relevant supporting experimental data.

Review of English language expression is however needed to correct some minor errors that will improve the presentation for readers.  In addition, some discussion regarding the protein's enzymatic activity should be included and some mention of how this activity may be involved in the presenting phenotype of the platelets.  Similarly, what is known of other similar oxidase proteins - is there a common enzymatic active site / motif and if so, can any other information be gleaned about the role of PCYOX1 in platelet or cell activity or in other cells for that matter. Some comment on these points should be included in the discussion.  Could any of this be used to speculate on the unknown component in plasma from KO Pcyox1 mice that is responsible for the changes seen in the platelet aggregation? Were any attempts made to compare protein profiles of the WT versus KO plasma?

Author Response

REVIEWER #1

The findings of this study strongly support a novel function for the protein Prenylcysteine Oxidase 1 in thrombosis involving platelet hypo-reactivity. The manuscript clearly outlines the strategy and rationale for the study. Its findings are clearly presented and argued, with relevant supporting experimental data.

We thank the Reviewer for his/her comments that we have real appreciated. We are grateful to him/her for the very important suggestions that have contributed to improve the manuscript. We have thoroughly revised the manuscript taking into account the points raised by the reviewer, and we have included this information in the new version.

  1. Review of English language expression is however needed to correct some minor errors that will improve the presentation for readers.

As suggested by the reviewer we performed an English revision.

  1. In addition, some discussion regarding the protein's enzymatic activity should be included and some mention of how this activity may be involved in the presenting phenotype of the platelets. Similarly, what is known of other similar oxidase proteins - is there a common enzymatic active site/motif and if so, can any other information be gleaned about the role of PCYOX1 in platelet or cell activity or in other cells for that matter. Some comments on these points should be included in the discussion. Could any of this be used to speculate on the unknown component in plasma from KO Pcyox1 mice that is responsible for the changes seen in the platelet aggregation?

We thank the Reviewer for these interesting issues. We have added comments in the Discussion section (pages 12-13; lines 252-260; 264-267; 278-280).

  1. Were any attempts made to compare protein profiles of the WT versus KO plasma?

We thank the Reviewer for this comment. As described in the Discussion section, we demonstrated that Pcyox1-/-/Apoe-/- mice have a significant decrease in plasma activity of PAI-1, the primary physiological inhibitor of fibrinolysis. This finding might explain, at least, the reduced thrombotic phenotype in Pcyox1-/-/Apoe-/- mice.

Reviewer 2 Report

In the paper named “Prenylcysteine Oxidase 1 (PCYOX1), a new player in thrombosis” author shown, in an animal model, that absence of Pcyoxy1 results in platelet hypo-reactivity and impaired arterial thrombosis and indicate Pcyoxy1 as a novel target for antithrombotic drugs. Moreover authors demonstrate recently that the secretome from PCYOX1-silenced cells reduced platelet adhesion both to fibrinogen and endothelial cells, suggesting a potential contribution of PCYOX1 into thrombus formation. The paper is well redacted and well structured and the results are clear presented.

Only minor changes are required

  • The figures are very small and are difficult to see
  • In paragraph between line 94-96 and in paragraph between line 205-207 some results are mentioned but a figure related to this data are missing
  • TRAP and ADP are only shown as abbreviations but nowhere is their meaning clarified or why these doses are used.
  • In the proteomic analysis by mass spectrometry only one transition is used to quantify, when usually more than 3 transitions were used in order to improve this quantification and even in most cases more than 1 peptide per protein was used. Why author used only one transition? Moreover only a transition was shown in Figure 4 but there are not quantitative results about these proteins. Have authors performed a MRM quantification? Can they give the quantitative values in platelets isolated from WT, WT plus TRAP and Pcyox1-/- and Pcyox1-/- plus TRAP ?
  • In figure 7 would be necessary a statistical analysis.
  • Some abbreviations in the text must be identified and a description must be provided as TRAP, ADP, PPP...
  • In Western blot section antibodies concentrations must be added.

Author Response

REVIEWR #2

In the paper named “Prenylcysteine Oxidase 1 (PCYOX1), a new player in thrombosis” author shown, in an animal model, that absence of Pcyoxy1 results in platelet hypo-reactivity and impaired arterial thrombosis and indicate Pcyoxy1 as a novel target for antithrombotic drugs. Moreover authors demonstrate recently that the secretome from PCYOX1-silenced cells reduced platelet adhesion both to fibrinogen and endothelial cells, suggesting a potential contribution of PCYOX1 into thrombus formation. The paper is well redacted and well structured and the results are clear presented.

We thank the Reviewer for his/her comments that we have real appreciated. We are grateful to him/her for the very important suggestions that have contributed to improve the manuscript. We have thoroughly revised the manuscript taking into account the points raised by the reviewer, and we have included this information in the new version.

Only minor changes are required

  1. The figures are very small and are difficult to see.

As suggested by the reviewer we enlarged the figures. We hope that in this way they could be now easier to see. We are ready to enlarge them again if necessary

  1. In paragraph between line 94-96 and in paragraph between line 205-207 some results are mentioned but a figure related to this data are missing.

In the first version, we decided to show data of TF activity and fibrinogen levels only in the main text of the manuscript. Now, as suggested by the reviewer, we also included the relative Figures (Figure 2 and Figure 10).

  1. TRAP and ADP are only shown as abbreviations but nowhere is their meaning clarified or why these doses are used.

We thank the reviewer for noticing this inaccuracy. According to the journal guidelines, we defined the meaning of these two acronyms the first time they appear in each section and figure of the manuscript.

The doses of platelet agonists were experimentally defined, identifying the optimal concentration to highlight a potential different platelets response between the two groups of mice analyzed. It was possible to pinpoint a significant difference in platelet aggregation only at sub-threshold concentrations of both agonists, while when the aggregation was maximum it was no more possible to appreciate a difference between WT and Pcyox1-/- mice, as expected.

  1. In the proteomic analysis by mass spectrometry only one transition is used to quantify, when usually more than 3 transitions were used in order to improve this quantification and even in most cases more than 1 peptide per protein was used. Why author used only one transition? Moreover only a transition was shown in Figure 4 but there are not quantitative results about these proteins. Have authors performed a MRM quantification? Can they give the quantitative values in platelets isolated from WT, WT plus TRAP and Pcyox1-/- and Pcyox1-/- plus TRAP ?

We agree with the reviewer that the use of more than 1 peptide could be more informative, but also commercially available kits, such as those traded by Cambridge Isotope Laboratories, provide only one peptide for the majority of the proteins. We have performed transition selection and optimization with Skyline tools using the heavy labelled peptide. Despite more than 2 transitions were detectable using high concentration of standards, the 2 selected transitions were the most intense and allowed us a lower limit of quantitation (0.125fmol/µl), also due to the increase in dwell time. We used the most intense as quantifier and the second as qualifier to confirm the retention time and the ratio dot product (rdotp > 0.96 calculated with skyline). We have tested the goodness of this approach also in cellular models with high levels of PCYOX1 expression in which we reduced the expression with shRNAs.

Nevertheless, according to the reviewer’s suggestion we have calculated the concentration using both the transitions as quantifiers. Figure 5 has been modified showing an image with the 2 transitions and the standard curve that we obtained with this approach. The method’s description has been changed accordingly. Concerning the last issue, we have planned future experiments to address the role of platelet PCYOX1 in response to different agonists.

.

  1. In figure 7 would be necessary a statistical analysis.

According to reviewer’s suggestion, results from statistical analysis have been added also in the figure (now Figure 8).

  1. Some abbreviations in the text must be identified and a description must be provided as TRAP, ADP, PPP...

We thank the reviewer for noticing this inaccuracy. According to the journal guidelines, we defined the meaning of all acronyms the first time they appear in each section and figure of the manuscript.

  1. In Western blot section antibodies concentrations must be added.

We apologize for this inaccuracy, we added this information in the Materials and Methods section, page 15 line 385